# Improvement of Adhesion Properties of Polyamide 6 and Polyoxymethylene-Copolymer by Atmospheric Cold Plasma Treatment

**DOI:** 10.3390/polym10121380

**Published:** 2018-12-12

**Authors:** Zoltán Károly, Gábor Kalácska, László Zsidai, Miklós Mohai, Szilvia Klébert

**Affiliations:** 1Institute of Materials and Environmental Chemistry, Research Centre for Natural Sciences Hungarian Academy of Sciences, Magyar tudósok krt. 2., H-1117 Budapest, Hungary; karoly.zoltan@ttk.mta.hu (Z.K.); mohai.miklos@mta.ttk.hu (M.M.); 2Institute for Mechanical Engineering Technology, Szent István University, Páter Károly u.1, H-2100 Gödöllő, Hungary; Kalacska.Gabor@gek.szie.hu (G.K.); Zsidai.Laszlo@gek.szie.hu (L.Z.)

**Keywords:** engineering polymers, cold plasma, DCSBD, adhesion, surface chemistry, XPS

## Abstract

A study is presented on cold plasma treatment of the surfaces of two engineering polymers, polyamide 6 (PA6) and polyoxymethylene (POM-C), by diffuse coplanar surface barrier discharges under atmospheric air conditions. We found that plasma treatment improved the adhesion of both polymers for either polymer/polymer or polymer/steel joints. However, the improved adhesion was selective for the investigated adhesive agents that were dissimilar for the two studied polymers. In addition, improvement was significantly higher for PA6 as compared to POM-C. The observed variation of the adhesion was discussed in terms of the changes in surface chemistry, wettability and topography of the polymer surface.

## 1. Introduction

Engineering polymers find broad application in any industrial area owing to their many favorable properties, such as machinability, dimensional stability, resistance to corrosion and chemicals combined with low cost. As reported in other research, the surface properties of polymers are also of paramount importance with regard to tribology and adhesion when they are joined with other materials or when they are coated or printed. From an adhesion point of view, factors promoting good adhesion include a large area of the contacting surfaces, high surface energy and good wettability [1,2]. Most polymers, however, possess poor surface properties, such as low surface energy and low wettability due to the insufficient polar groups on the surface. The improvement of surface properties has been the goal of earlier research [3,4,5,6,7]. Among the numerous attempts to improve the surface properties, cold plasma treatments have become increasingly popular. This is due to environmental concerns [8] and because cold plasma treatment subsequently provides a viable technical solution under atmospheric pressure at a relatively low cost [9,10,11]. Plasma treatment improves the adhesive properties of polymers by inducing several changes on the polymer surface—the ionized species and free radicals present in the plasma remove organic impurities commonly present on the surface. The surface becomes more activated by the formation of polar functional groups such as hydroxyl, carbonyl or carboxylic acid [12,13]. As a result, considerable changes take place in the chemical composition of the topmost layer and, consequently, in the surface energy of the treated polymer. In addition, plasma treatment is also capable of exerting an etching effect on the surface, leading to a modified morphology and increased surface roughness [14]. While the cold plasma treatment of various polymers such as polyethylene, polypropylene, polyether ether ketone, polyethylene terephthalate, polytetrafluoroethylene, etc. have been extensively investigated [15,16,17], rather limited publication is available on some engineering polymers [18,19,20]. The goal of this paper is to study the effects of diffuse coplanar surface barrier discharge (DCSBD) atmospheric pressure plasma on surface modification and consequent properties of polyamide 6 (PA6) and polyoxymethylene (POM-C). In particular, this paper investigates how adhesive properties are related to the changes in surface chemistry, wettability and surface roughness. 

## 2. Material and Methods

### 2.1. Materials and Sample Preparation

Virgin polyamide 6 (PA6) and polyoxymethylene (POM-C) used in this study was distributed by Quattroplast Ltd., Budapest, Hungary and produced by Ensinger GmbH, Nufringen, Germany. Selected characteristic properties of the test materials are listed in Table 1. For the adhesive tests, rectangular specimens with dimensions of 25.4 mm × 100 mm × 2 mm were cut from commercial grade extruded plates using a metal saw disc. A long strip was cut parallel to the longer edge of the plate in a width of 25.4 mm, followed by slicing the strip into the final size. The surfaces used for adhesion tests were first polished with silicon carbide abrasive paper (grit numbers P1200 and 2000) under wet conditions, followed by dry polishing with a felt sheet. Subsequently, the samples were rinsed with distilled water and ethanol. A standard steel grade S235 (Ferroglobus Ltd., Budapest, Hungary) was used as a counterface in the adhesion tests. The steel surfaces were also polished with SiC abrasive paper (grit numbers 400 and 600) to a mean surface roughness of Ra = 0.07 ± 0.02 µm. Finally, the surface was chemically cleaned from organic contaminants. 

### 2.2. Plasma Treatment

A diffuse coplanar surface barrier discharge (DCSBD) (manufactured by Roplass s.r.o., Brno, Czech Republic) plasma system was employed for surface activation [21,22]. The plasma panel consisted of two systems of parallel strip-like electrodes (with typical dimensions of 1.5 mm wide, 0.5 mm thick, 1 mm strip to strip) embedded in an aluminum oxide matrix (Figure 1). The ceramic layer between electrodes and the plasma had a thickness of typically 0.4 mm. The plasma was ignited with a high frequency (10–20 kHz), high voltage with peak-to-peak values of 20 kV. The elementary discharge comprised a diffuse surface discharge developing over the metal electrodes and a filamentary streamer discharge created between the electrodes, giving its H shape. With increasing voltage and absorbed power as more and more elementary discharges are generated, visually homogenous plasma can be reached. The high voltage applied may also lead to the heating of the dielectric surface and the surrounding gas. Oil was circulated over the system to keep the system at the lowest possible temperature. It also kept the gas temperature around 320 K. The DCSBD plasma system was operated at a power of 320 W, which provided a quasi-homogeneous diffuse plasma with air as the process gas. 

In the plasma tests, one side of the polymer samples was treated with DCSBD plasma for 30 s. A constant distance of 0.5 mm was maintained between the DCSBD surface and the polymer specimen. The latter was gently moved back and forth within a 1-cm period distance along the DCSBD surface so as to provide an even more homogeneous diffuse plasma contact. 

### 2.3. Adhesive Testing

For adhesive testing, lap-shear tests were carried out according to DIN EN 1465 on single lap joints of polymer/polymer or polymer/steel pairs with a bonded area of 25.4 mm × 12.5 mm. Three commercial adhesives were employed including a cyanoacrylate (Loctite 406) and an acrylic based adhesive (Loctite 3035, two-component glue: tetrahydrofurfuryl methacrylate, alkyl methacrylate, organoboron amine) as well as a two-component epoxy adhesive (Loctite 9466) from the same manufacturer (Henkel AG & Co., Düsseldorf, Germany). The adhesives were applied onto the bonded area with a controlled thickness of 0.1 mm. Prior to using the cyanoacrylate type adhesive primer, Loctite 770 was applied on the polymer surface according to the manufacturer’s recommendation. During curing, a constant normal load of 5 N was maintained for 24 h over the bonded area. The plasma-treated specimens were glued right after treatment and stored in a plastic box until testing. The pulling tests were performed using a universal mechanical tensile bench (Zwick Roell Z100, max. 100 kN) with a pulling speed of 1.3 mm/min. The maximum load upon failure with respect to the bonded surface area was used to calculate the shear strength of the adhesive bond.

### 2.4. Surface Characterization

The chemical, morphological and energetic properties of the plasma-treated surface were determined by X-ray photoelectron spectroscopy (XPS), sessile drop contact angle measurements and three-dimensional (3D) non-contact optical profilometry (CCI Optics). 

Contact angles measurements were performed by the static sessile drop method at room temperature using double distilled water and diiodomethane (Sigma–Aldrich, Budapest, Hungary, Reagent Plus 99% grade), applying the SEE System apparatus (Advex Instruments, Brno, Czech Republic). The measuring liquids were deposited in 2-µl droplets by Hamilton syringe. The calculated contact angle values were presented as the average of five consecutive measurements, performed always on previously non-wetted parts of the samples. The surface free energy with its polar and dispersive components was calculated following the Owens–Wendt method [23].

X-ray photoelectron spectra were recorded on a Kratos XSAM 800 spectrometer operating in fixed analyzer transmission mode, using Mg Kα_1,2_ (1253.6 eV) excitation. Survey spectra were recorded in the kinetic energy range of 150–1300 eV with 0.5-eV steps. Photoelectron lines of the main constituent elements, like the O1s, N1s and C1s, were recorded by 0.1-eV steps. The spectra were referenced to the C1s line (binding energy, BE = 285.0 eV) of the hydrocarbon type carbon. A Gaussian-Lorenzian peak shape (70/30 ratio) was used for peak decomposition. Quantitative analysis, based on peak area intensities after the removal of the Shirley-type background, was performed by the Kratos Vision 2 and XPS MultiQuant programs [24], using the experimentally determined photo-ionization cross-section data of Evans et al. [25] and the asymmetry parameters of Reilman et al. [26]. Surface chemical compositions were calculated by the conventional infinitely thick layer model, where all components are supposed to be homogeneously distributed within the sampling depth detected by XPS.

Surface roughness and topographic analysis were investigated by non-contact profilometry, using a 3D surface optical profilometer Coherence Correlation Interferometry (CCI) HD type (Taylor Hobson) with an ultra-high precision closed loop piezoless z-scanner having a resolution in the z-direction of 0.1 Å. A scanned area of 350 × 350 µm^2^ was imaged for all specimens by vertical scanning interferometry, with an objective lens at 50× magnification. The images were evaluated by Talymap software (Digiserve) to calculate the 3D surface roughness parameters including Sa (average roughness) and Sz (maximum height). Roughness values were presented as the average of three measurements at independent surface locations, with repeatability Sa < 0.2 Å.

## 3. Results and Discussion

### 3.1. Contact Angle Measurements

In Table 2 we list the contact angle values of water and diiodomethane. The surface energy values were calculated including both the polar and dispersive components for the pristine and plasma-treated samples. The investigated polymers were found to be rather hydrophobic, with their high-water contact angle values being 70° and 73° for PA6 and POM-C, respectively. Even a 30-second DCSBD plasma treatment could greatly decrease the contact angles down to ca. 28° and 41° for PA6 and POM-C, respectively. It was found that longer treatment time did not significantly improve the wettability. A considerable decrease in the water contact angle values was found for a range of polymers after cold plasma treatment in several research papers, as compared by Dixon [16]. Parallel to the decrease of water contact angles, the surface energies increased mainly owing to the rise in the polar component. Since the longer plasma treatment did not result in higher surface energies, all other tests were performed on specimens exposed to plasma treatment for only 30 seconds. The effect of plasma treatment, however, is reported to vanish after a longer period for plasma-treated polymers due to the surface reactivity and reorientation of polymer chains. We also found a slight drop in the surface energy for both polymers even one day after treatment. However, the value did not decrease further significantly and it was practically maintained even after ca. 3 months of the treatment (Figure 2). From an application point of view, the obtained results suggest that any planned processing step such as functionalizing, dyeing or applying adhesives on the plasma-activated polymers can be expanded in time after plasma treatment. 

### 3.2. Analysis of Surface Chemistry after Plasma Treatment

The changes in the surface chemical composition of the studied polymers on plasma activation were monitored by XPS analysis. The elemental compositions (atomic %) of the polymer surfaces before and after plasma treatment calculated from the survey spectra are presented in Table 3. High resolution XPS spectra of the C1 line were used to determine the quality and quantity of the developed polar groups on the surface. The assignment of the peak components to chemical features is also summarized in Table 4.

The broad and asymmetric shape of the C1 envelopes could be decomposed into several components for both polymers (Figure 3). The C1 component, which is attributed to C–C and C–H bonding states (predominant peak at 285.0 eV), has a much higher quantity on the surface of the pristine specimens than the theoretical composition. This is the result of the adventitious carbon contamination of the surface, which is typical for all samples stored under ambient conditions. Correction can be applied for this adventitious carbon [27]. The corrected values are also shown in Table 4 for the pristine polymers. The corrected compositions are in good agreement with the expected values. The thickness of the contamination layer can also be calculated by the layers-on-plane model, which was confirmed to be ca. 1 nm for both PA6 and POM-C. Upon treatment with DCSBD plasma, the oxygen content increased parallel to the carbon content, indicated by the changes of O/C atomic ratios for both polymers. In particular, for the DCSBD-treated PA6, the relative intensity of the C1 component significantly decreased, while that of the C2, C3, and C4 components exhibited a slight increase. In addition, a new component (C5) appears, which can be assigned to urethane-like group (N–(C=O)–O). In the N1s spectrum the intensity of N1 component (amide) decreases, while a new component appears (urethane). In the O1s region, O1 decreases and new components also appear at 533.1 eV, indicating the formation of carboxyl groups (Figure 3). The increased amount of polar groups incorporated onto the surface layer can explain the better wettability. These results agree with findings of earlier research [28,29], even though surface modification was performed by a different cold plasma method.

For the DCSBD-treated POM-C, in the C1s region, the intensity of C1 component significantly decreased, but other changes could not be detected (Figure 4). This suggests that the main effect of the DCSBD plasma treatment for POM-C was to practically eliminate the surface hydrocarbon-type contaminations. Since molecules of POM-C are intrinsically highly polar, the treated polymer exhibited better wettability as compared to the pristine specimen. As long as recontamination of the surface can be prevented, the wettability can be retained.

### 3.3. Surface Topography

3D surface scans using non-contact profilometry were performed to monitor the surface topography of pristine and plasma-treated samples. We compared the 3D surface roughness parameters for Sa (average roughness) and Sz (maximum height) before and after plasma surface treatment in Figure 5. 

The two polymers greatly differed in their initial roughness values (Sa), being 0.2 µm for POM-C and 1.35 µm for PA6, despite the similar preparation method before plasma treatment. We attribute the difference in the roughness to the dissimilar surface hardness and strain capability of the polymers during preparation (machining and abrasive polishing). Plasma treatment, however, decreased the roughness for both polymers, although the effect was much higher for PA6. The roughness Sz of the originally polished surface decreased due to the melting and smoothing of the asperities. As the asperities were much higher for the pristine PA6, it was affected to a greater extent as compared to POM-C. The DBD Dielectric Barrier Discharge (DBD) plasma commonly increases the surface roughness of polymers due to etching effects [30,31,32], which is more pronounced for the amorphous regions [33]. However, when the treated polymer possesses increased roughness for any reason, such as prior polishing, the smoothing effect of the plasma seems to be decisive.

### 3.4. Adhesive Tests

Figure 6a,b show the shear strength of pristine and plasma-treated polymer/polymer and polymer/steel joints evaluated by lap-shear tests for PA6 and POM-C, respectively. By applying the adhesives on the untreated PA6 surfaces, the obtained shear strength was 10%–35% of the tensile strength of the polymer. The shear strength of the polymer/polymer joints of PA6 using the epoxy adhesive (Loctite 9466) was only 10% of the tensile strength. All the adhesives became rigid after setting and peeled off the surface from the PA6 polymer that is capable of significant strain. Following DCSBD treatment, the adhesion strength improved for each type of adhesive. The highest improvement both for the polymer/polymer and polymer/steel joints occurred when acrylic adhesive (Loctite 330) was used. In the former case, the obtained shear strength was close to the tensile strength of the pure polymer. The improvement of the adhesion can be expected from the better wettability and higher surface polarity of the plasma-treated polymer surfaces. In addition to the increased adhesion strength, the reliability of the joints greatly improved. This was indicated by the decrease of the statistical deviation of the shear strength (five repetitions) after plasma treatment for all adhesives and joint types, from about 9% for the pristine samples to 3% on average for the plasma-treated PA6. 

The adhesion of pristine POM-C was rather poor for any type of adhesive and joints relation. The adhesives detached from the surface easily, implying a lack of strong bonds between the polymer surface and the adhesive. The obtained shear strength was a meager 10%–20% of the tensile strength of the polymer. Following DCSBD treatment, the adhesion did not change much for most adhesives except the two-component epoxy one (Loctite 9466). This latter showed significant improvement in the adhesion both for polymer/polymer and polymer/steel joints. However, even in the case of this improved adhesion, the shear strength remained below the tensile strength of POM-C. Comparing the sharply different adhesion behaviors of the two investigated polymers after DCSBD treatment, several reasons can be found for the better performance of PA6. Firstly, DCSBD treatment created extra oxygen-containing polar moieties on the surface of PA6, while for POM-C the surface was simply cleaned from the hydrocarbon contamination film and no new functional groups were formed. While the surface roughness decreased with DCSBD treatment for both polymers, the mean roughness of DCSBD-treated PA6 remained three times that of POM-C. Moreover, much higher maximum values with regard to the highest peaks were observed, which ultimately gave rise to a higher contact area available for the adhesives. The higher polarity and roughness of the surface increased the surface energy of PA6 by ca.10 mJ/m^2^ above that of POM-C. 

With respect to the different joints and polymers, the observed failure could be mainly attributed to adhesive failure on the surface of the polymer or steel. Cohesive failure within the adhesive film or cracking in the bulk polymer could not be detected. For the pristine PA6, the adhesives detached from the polymer surface. However, after DCSBD treatment for polymer/steel joint relations, the adhesives peeled off the steel surface, indicating such a stronger bond to the polymer. For pristine POM-C adhesive, failure occurred primarily on the polymer surface that did not change after treatment, but failure occurred at a generally higher applied shear stress. 

## 4. Summary

The atmospheric air dielectric coplanar surface barrier discharge (DCSBD) plasma treatment created significant physical and chemical changes on the topmost surface of the investigated engineering polymers (PA6 and POM-C). This ultimately had a significant effect on the adhesion properties as well. While the DCSBD treatment resulted in new oxygen-containing functionalities on the surface of PA6, according to XPS analysis for POM-C only, the carbohydrate contaminated layer was removed. The increased number of the available polar groups on the topmost surface increased the surface energy, as well as improved wettability after treatment. DCSBD plasma treatment had considerable benefit for the adhesion properties for both polymers. However, the increased adhesion could only be observed for one type of adhesive. For PA6 it was the acrylic (Loctite 330), while for POM-C it was the epoxy-based adhesive (Loctite 9466) that exhibited improved adhesion both in polymer/polymer and polymer/steel joint relations. For PA6, not only the adhesive shear strength increased but the DCSBD treatment also improved the reliability of the joints, as indicated by the much smaller standard deviation of the measured values.

## Figures and Tables

**Figure 1 polymers-10-01380-f001:**
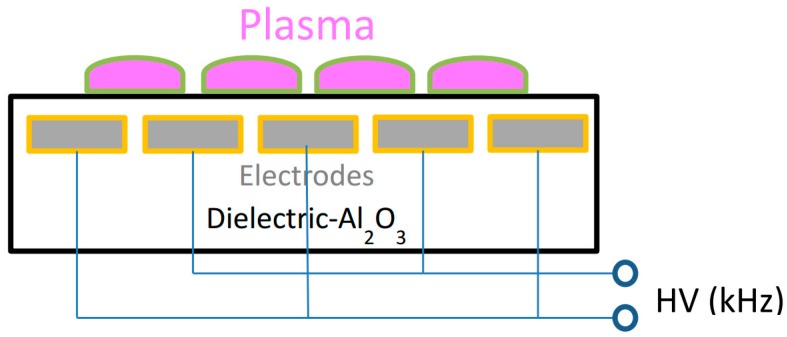
Sketch of the experimental diffuse coplanar surface barrier discharge (DCSBD) plasma panel.

**Figure 2 polymers-10-01380-f002:**
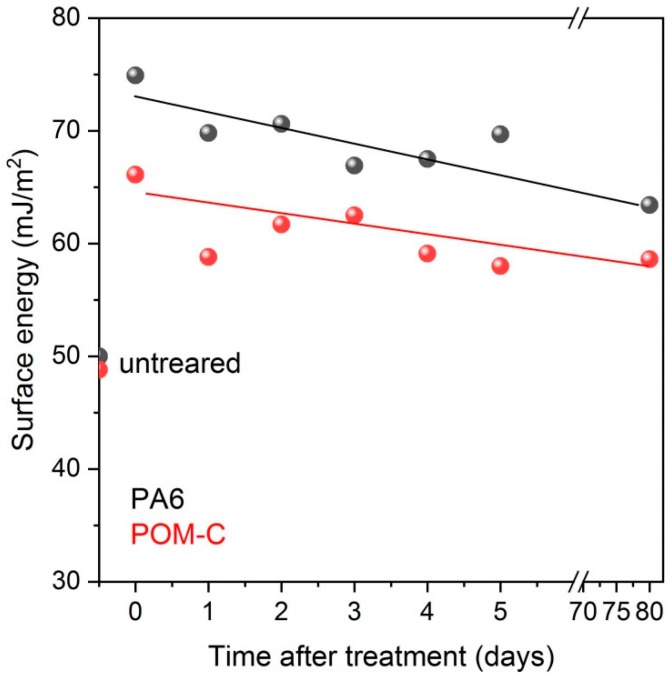
Surface energy values of the studied polymers after DCSBD plasma treatment as a function of time.

**Figure 3 polymers-10-01380-f003:**
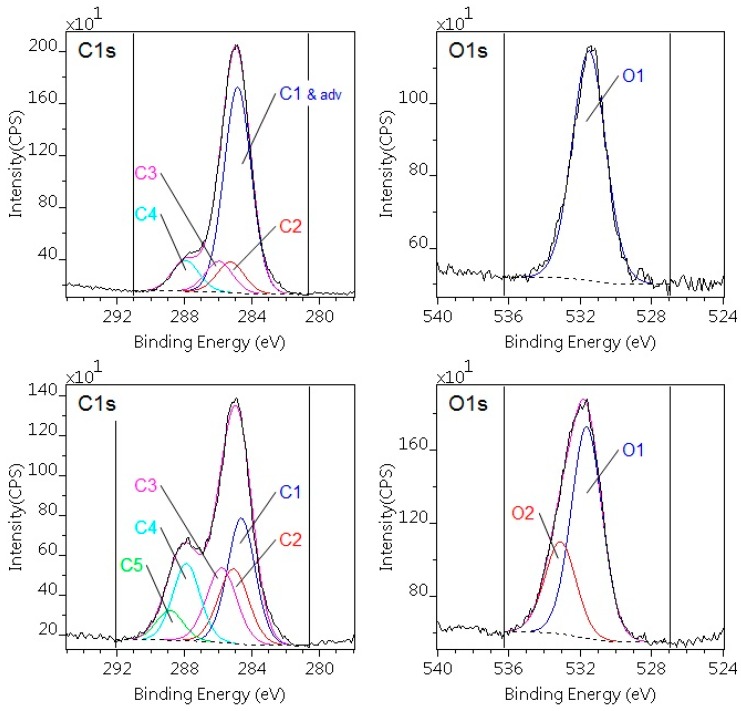
C1s and O1s photoelectron spectra of the untreated (upper) and plasma-treated (lower) PA6 sample. On the untreated sample the C1 component coincides with the adventitious carbon.

**Figure 4 polymers-10-01380-f004:**
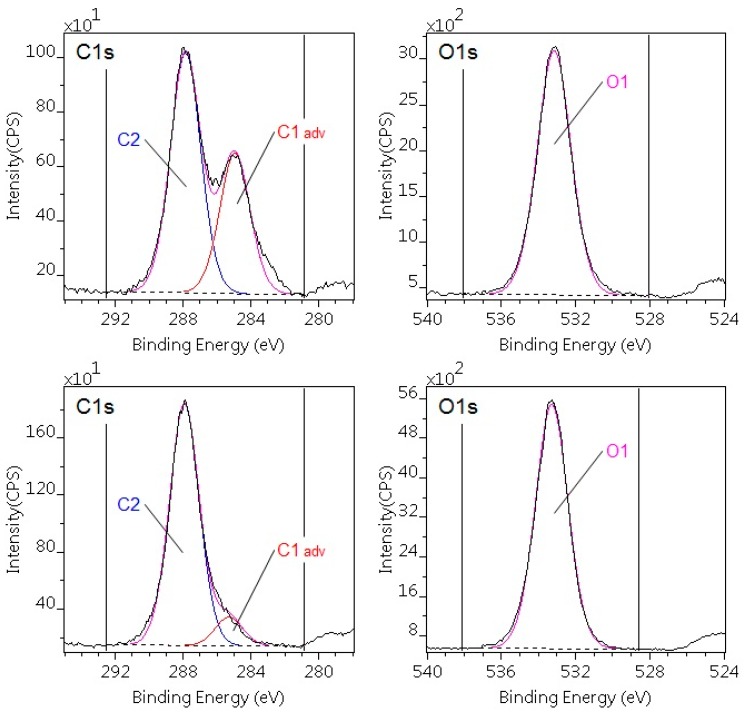
C1s and O1s photoelectron spectra of the untreated (upper) and plasma-treated (lower) POM-C sample. The treatment reduced the C1 adventitious carbon component.

**Figure 5 polymers-10-01380-f005:**
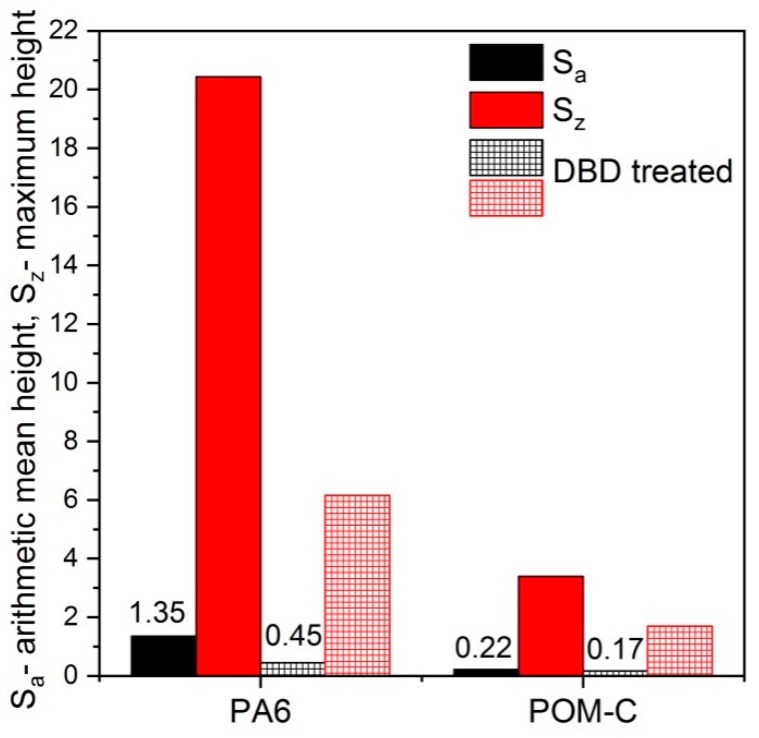
Major surface roughness parameters of pristine and DCSBD plasma-treated surfaces for PA6 and POM-C.

**Figure 6 polymers-10-01380-f006:**
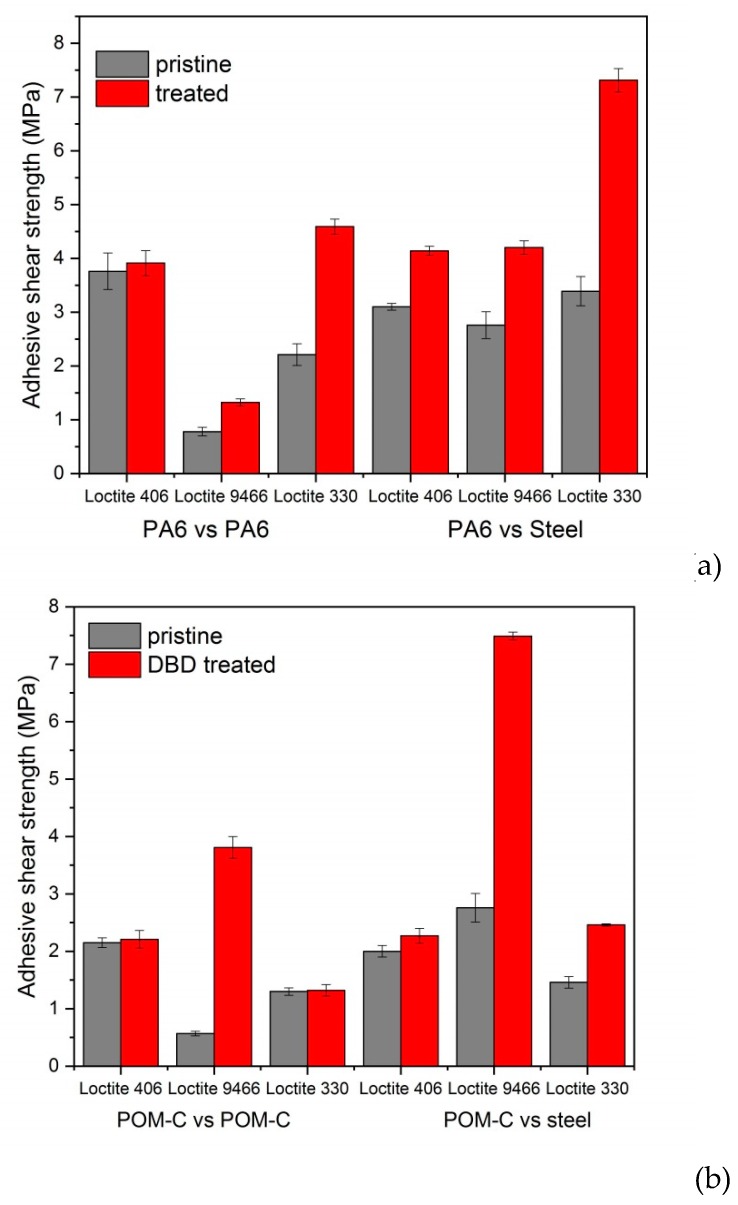
Adhesive shear strength of pristine (grey) and plasma-treated (red) polymer/polymer and polymer/steel joints using various adhesives for PA6 (**a**) and POM-C (**b**).

**Table 1 polymers-10-01380-t001:** Characteristic properties of testing materials.

Property	Polyamide 6 (PA6)	Polyoxymethylene (POM-C)
Density (g/cm^3^)	1.14	1.4
Yield stress (MPa)	70	65
Elasticity modulus (N/mm^2^, MPa)	3100	2700
Shore D hardness	75	85
Thermal conductivity (W/mK)	0.23	0.31
Melting temperature (°C)	255	170

**Table 2 polymers-10-01380-t002:** Contact angle values and surface energy values of the pristine polymers and plasma-treated samples measured at certain times after treatment.

Sample	θ_w_ (°)	θ_CH2l2_ (°)	γ_pol_ (mJ/m^2^)	γ_disp_ (mJ/m^2^)	γ_tot_ (mJ/m^2^)
PA6 pristine	70 ± 7.2	32 ± 2.1	6.3	43.6	50.0
PA6 treated, 30 s	28 ± 4.1	25 ± 2.6	26.1	46.2	72.3
PA6 treated, 60 s	26 ± 1.0	28 ± 2.3	27.6	45.0	72.6
PA6 treated, 180 s	21 ± 2.4	26 ± 2.7	29.1	45.8	74.9
POM-C pristine	73 ± 4.2	32 ± 2.1	5.2	43.6	48.8
POM-C treated, 30 s	41 ± 4.6	16 ± 3.6	18.9	48.9	67.8
POM-C treated, 60 s	44 ± 5.5	20 ± 1.3	17.8	47.9	65.8
POM-C treated, 180 s	43 ± 4.2	20 ± 3.2	18.2	48.0	66.1

**Table 3 polymers-10-01380-t003:** Surface composition (atomic %) of the pristine and plasma-treated polymer samples determined by X-ray photoelectron spectroscopy (XPS).

Sample	C	C CO	O	N	O/C ratio
PA6 pristine	82.6	-	8.8	8.6	
Corrected	73.8	-	12.8	12.5	0.17
PA6 treated	67.3	-	20.3	13.1	0.30
POM-C pristine	22.5	38.4	39.0	-	1.01
POM-C treated	5.8	47.7	46.5	-	0.97

**Table 4 polymers-10-01380-t004:** The results of quantification and peak assignment of C1 components of the pristine polymers and plasma-treated samples by XPS analysis.

C components	Composition (atomic %)			Binding energy (eV)	Chemical states
	Pristine	Contamination-corrected pristine	Plasma-treated		
PA6					
C1	57.1	36.6	21.1	285.0	C–C, C–H
C2	8.5	12.4	13.8	285.3	CH–C=O
C3	8.5	12.4	13.8	286.0	C=O, C–N
C4	8.5	12.4	16.3	288.0	C=O, N–C=O
C5	0.0	0.0	2.4	289.3	O=C–O(H)
POM-C					
C1	22.5	—	5.8	285.0	C–C, C–H
C2	38.4	47.7	47.7	287.9	O–C–O

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
