# Peer review of "Improvement of Adhesion Properties of Polyamide 6 and Polyoxymethylene-Copolymer by Atmospheric Cold Plasma Treatment"

_polymers, 2018, doi:10.3390/polym10121380_

Reviewer 1 Report

The paper reports on effect of adhesive performance increasing by dielectric coplanar surface barrier discharge plasma treatment for polymer/polymer and polymer/steel joints. The paper shows a complete experimental research, suitably carried out and well described. The experimental process, analysis and results are clearly shown and explained. Based on the current status of the manuscript, I would recommend its publication in Polymers with minor revisions.

 Additional comments are as the followings:

1.      In chapter 2.2 Plasma treatment, the experimental setup for dielectric coplanar surface barrier discharge plasma is well explained. However, to increasing understandings for experimental setup, the authors are encouraged to insert related figures such as the schematic diagram or experimental figure.

2.      In table 2., the contact angles and surface energy values are presented without any relations or explanations. In this point, the relations between contact angles and surface energies should be presented.

3.      In fig. 1, the data is indicated only as point values. However to show the effect of surface energy relation by time after treatment clearly, a tendency line should be inserted for each cases.

Author Response

Thank you for your comments and critical remarks. We have now corrected and improved the manuscript according to your advice. Please find below our answers to your comments:

Comment: In chapter 2.2 Plasma treatment, the experimental setup for dielectric coplanar surface barrier discharge plasma is well explained. However, to increasing understandings for experimental setup, the authors are encouraged to insert related figures such as the schematic diagram or experimental figure.

A: A schematic diagram of the experimental DCSBD device is inserted as Fig. 1.

Q: In table 2., the contact angles and surface energy values are presented without any relations or explanations. In this point, the relations between contact angles and surface energies should be presented.

A: It is commonly accepted that surface energy can be determined using the values obtained from sessile drop measurements. For calculations several theories are available but for polymers with typically polar surfaces the Owens/Wendt theory is widely applied. I do not think that the related equations should be detailed in the paper as it has less importance with respect to the subject of our research. However, we certainly referred the method [23] in the experimental part (Section 2.3, end of the second paragraph).

Q: In fig. 1, the data is indicated only as point values. However to show the effect of surface energy relation by time after treatment clearly, a tendency line should be inserted for each cases.

A: A tendency line has been inserted for each polymer.

Reviewer 2 Report

The manuscript is well written. The topic is interesting. The obtained results bring the advancement into the studied subject.

However, there would be great some minor improvement:

1: Table 2: There is clearly visible, that all surface modification is done before the duration of the treatment of 30 s, when is almost constant with another treatment. There would be important to enrich the table with the values for 5 and 10 s.

2: Fig. 5 a-b: Statistical analysis is missing. 

Author Response

Thank you for your comments and critical remarks. We have now corrected the manuscript according to your advice.Please find below our answers to your comments:

Comment: Table 2: There is clearly visible, that all surface modification is done before the duration of the treatment of 30 s, when is almost constant with another treatment. There would be important to enrich the table with the values for 5 and 10 s.

A: Indeed, surface modification is almost completely finished within 30 s treatment time. While the suggested 5 and 10 s plasma treatment would be interesting, it could be hardly carried out precisely with our experimental set-up because it takes itself a few seconds to reach the “standard” experimental operating conditions in every respect and the uncertainty of the treatment time for either 5 or 10 seconds would be too high regarding a scientific paper. Anyway, I think the optimization of treatment time is rather a technical issue than scientific.

Q: Fig. 5 a-b: Statistical analysis is missing.
A: You are right, we forgot to indicate the deviations of the obtained values on the diagram, although it is really important as we speak about it. The diagram has now been corrected.